# Canopy Gaps Control Litter Decomposition and Nutrient Release in Subtropical Forests

**Jiajia Chen** [1,2], **Jing Zhu** [1,2], **Ziwei Wang** [2,3], **Cong Xing** [1,2], **Bo Chen** [1,2], **Xuelin Wang** [1,2], **Chensi Wei** [1,2], **Jinfu Liu** [1,2,3], **Zhongsheng He** [1,2] and **Daowei Xu** [1,2,*]

1. Key Laboratory of Fujian Universities for Ecology and Resource Statistics, College of Forestry, Fujian Agriculture and Forestry University, Fuzhou 350002, China
2. Cross-Strait Nature Reserve Research Center, Fujian Agriculture and Forestry University, Fuzhou 350002, China
3. College of Computer and Information Sciences, Fujian Agriculture and Forestry University, Fuzhou 350002, China
* Correspondence: xudaowei2004446@126.com; Tel.: +86-137-9942-8792

**Abstract:** The formation of a canopy gap results in changes to the microenvironment which, in turn, affect litter decomposition and nutrient release. However, the mechanisms underlying these effects in differently sized gaps and non-gaps remain poorly understood. To address this gap in knowledge, we selected three large gaps (above 150 m$^2$), three medium gaps (50–100 m$^2$), three small gaps (30–50 m$^2$), and three non-gaps with basically the same site conditions. We then used the litter bag method to investigate leaf and branch litter decomposition over a year in a *Castanopsis kawakamii* natural forest with the aim of characterizing the litter mass remaining and the nutrient release in canopy gaps and non-gaps. Our results revealed that the remaining litter mass of leaf and branch litter was lower in medium gaps compared to other gaps, and leaf litter decomposed faster than branch litter. Environmental factors were identified as the primary drivers of total carbon and nitrogen release during litter decomposition. Gap size (canopy openness), taxonomic Margalef index, the Brillouin index of soil microbes, soil total nitrogen content, soil pH value, and average air temperature were identified as the main factors driving carbon and nitrogen release from branch litter. In the late decomposition stage, the taxonomic Pielou index, soil total potassium content, soil water content, and average relative air humidity were the main drivers of nutrient release from branch litter. The soil water content and average relative air humidity were also found to be the main factors affecting the nutrient release from leaf litter throughout the different stages of decomposition. Overall, our study highlights the impact of canopy gaps on microenvironmental variation, taxonomic community diversity, and soil microbial functional diversity and how these factors ultimately influence litter decomposition and nutrient release. Our findings provide an important foundation for further research into soil nutrient cycling in subtropical natural forests.

**Keywords:** canopy gap; litter decomposition; nutrient release; taxonomic diversity; soil microbial diversity

## 1. Introduction

Litter is the key bridge for biomass circulation and energy flow between the aboveground and underground components of forests [1], and it is the basic carrier of the carbon pool and nutrients in forest ecosystems [2]. Therefore, it is commonly acknowledged that the process of litter decomposition drives many critical functions of terrestrial ecosystems [3]. Under the comprehensive effects of its own characteristics, the climate, taxonomic diversity, and soil microbial communities [4,5], litter returns nutrients to the soil by decomposing and releasing organic matter. These effects directly improve soil fertility, alter the species composition of the forest community, and influence the community succession [6]. The stoichiometric characteristics of plants, litter, and soil are the key indicators of forest nutrient limitation and cycling in response to climate change; meanwhile, their interactions jointly affect forest nutrient cycling and regulation strategies [7]. Thus, by revealing the

influencing factors of litter decomposition, we can further understand the processes of nutrient cycling and organic matter turnover in forest ecosystems.

Litter quality (chemical and physical composition), decomposer (soil microbial communities), and environmental factors are regarded as the predominant drivers of litter decomposition [8,9]. Plant taxonomic diversity alters the litter composition (mixed litter) and chemical characteristics, increases resource heterogeneity, and changes the abundance of decomposer communities, accelerating the litter decomposition rate compared with monospecific litter [10]. The soil microbial community, as the main driver of nutrient cycling in forest soil [11], plays a critical role in litter decomposition [12]. With variations in environmental conditions, the quality and abundance of the soil microbial community may affect the litter decomposition rate [4]. Meanwhile, the feedback mechanism between the taxonomic diversity and soil microbial community stimulates the decomposition rate. In turn, plant roots absorb and utilize the nutrients released by litter, and the roots accelerate litter decomposition to modulate the cycling process of forest soil ecosystems [6,13]. A high taxonomic diversity alters the type and quantity of root exudates, and the secondary metabolites secreted by the roots stimulate soil chemical properties or microbial activities which indirectly affect litter decomposition [14,15].

Canopy gaps serve as carriers of environmental heterogeneity, and their formation enhances the availability of light, temperature, precipitation, and other resources, impacting the physical and chemical properties of the soil. As a result, they influence the taxonomic diversity, soil microbial diversity, and litter stoichiometric characteristics of forests [16–18]. Canopy gap sizes can also increase the leaching effect of precipitation on the litter surface and regulate the processes of litter mass remaining and nutrient release [19,20]. Hence, litter decomposition may be co-driven by the environmental factors, taxonomic diversity, and the soil microbial communities of canopy gaps. Understanding the relative importance of these drivers of canopy gaps to litter decomposition will further elucidate the process of the utilization of soil nutrients in forest ecosystems.

An evergreen broad-leaved forest is a unique type of forest vegetation that has its widest distribution and highest biodiversity in subtropical forests. *Castanopsis* is the dominant group in subtropical evergreen broad-leaved forests, and it plays an important role in the ecological stability of evergreen broad-leaved forests [21,22]. Nutrient release from the decomposition of *Castanopsis* litter directly affects the nutrient cycle of the subtropical forest soil ecosystem. Sanming *Castanopsis kawakamii* Nature Reserve is a typical subtropical forest dominated by *C. kawakamii* tree species, representing an ideal place to study litter composition in subtropical forests. This natural forest is mostly composed of hundred-year-old trees, and the number of forest gaps increases due to the serious fragmentation of the canopy, leading to changes in the understory microenvironment. It provides a good research platform for the study of forest litter decomposition. However, the mechanisms of the effects of canopy gaps, taxonomic diversity, and soil microorganisms on litter decomposition in canopy gaps are still unclear, limiting the understanding of the material cycle process of forest ecosystems.

Two methods are usually used for litter decomposition research: the litter bag method, in which a known quantity of litter is enclosed in a bag made of nylon or other inert material, and the leaf-pack method, in which litter is fastened together with plastic buttoneers or a monofilament fishing line [23]. In a comparison between the litter bag method and the leaf-pack method, litter bags provide an artificial substratum for macroinvertebrates which is difficult to correct, are prone to vandalism [24], and are more expensive to construct than leaf packs. However, the conditions within litter bags approximate those occurring outside, any organisms that do grow can be easily and reproducibly removed, and the materials do not escape through the mesh [23,25]. Therefore, we selected the litter bag method for this study. We then elucidate the effects of canopy gaps of different sizes on litter decomposition and nutrient release and reveal the main factors that influence variation during the different stages of decomposition. We assume that the various characteristics relating to the litter mass remaining and the nutrient release from leaf and branch litter in different sizes of

canopy gaps are different. We hypothesize the following: (1) the litter mass remaining and nutrient release in different canopy gap sizes are asynchronous; (2) compared with taxonomic diversity and soil microbial diversity, the heterogeneity of environmental factors caused by gap formation may be the main factor driving litter decomposition.

## 2. Materials and Methods

### 2.1. Study Site

This study was conducted at the *Castanopsis kawakamii* Nature Reserve (117°27′09″–117°30′02″ E, 26°10′02″–26°12′43″ N) in a subtropical forest of China. The elevation of this reserve varies between 180 and 604 m. The study site has a subtropical monsoon climate with an average annual temperature of 19.5 °C, an average annual precipitation of approximately 1500 mm, and an average annual relative humidity and wind speed of 79% and 1.6 m/s, respectively. The soil type of this forest mainly consists of acidic ferro aluminate with abundant humus, which is rich in soil nutrition [20].

In April 2018, we used an unmanned aerial vehicle (UAV DJI Phantom 4 Pro, SZ DJI Technology Co., Ltd., Shenzhen, China) equipped with a digital camera (X3–FC350, Nikon, Tokyo, Japan) to obtain high-resolution remote sensing images of the region. We used a Nikon D7200 with a fisheye lens (NIKON DX AF FISHEYE NIKKKOR 10.5 mm 1:2.8G ED, Nikon, Japan) to capture an orthogonal image of the canopy gap. Meanwhile, we used the two hemispherical photographs (THP) method to calculate the canopy gap size, and the camera shooting distance was one meter from the ground [26]. According to the canopy gap size and the field survey, we selected three large gaps (above 150 m$^2$), three medium gaps (50–100 m$^2$), and three small gaps (30–50 m$^2$) with basically the same site conditions. Simultaneously, we also set three non-gaps with dimensions of 10 m $\times$ 10 m at a distance of 10 m from the canopy gap. We used the Gap Light Analyzer software to calculate the canopy openness [27], and recorded the topographic factors (e.g., longitude, latitude, elevation, slope, slope position, and slope direction) of the canopy gaps and non-gaps. The crown deviation was calculated by the ratio of the maximum branch length of the plant in the canopy gap to the crown width at the connection. Detailed information of the sample plots is shown in Table A1.

### 2.2. Experimental Design

In April 2018, we set five litter traps of 1 $\times$ 1 m$^2$ at a distance of 1 m above the soil surface in the east, south, west, north, and center position of each canopy gap and non-gap to regularly collect the fresh litter that had not decomposed [28]. The litter on undisturbed forest floors typically consists of multiple species which, in turn, can drive interactions among microbes and invertebrates [29]. Since *C. kawakamii* is the dominant tree species in this natural forest, the litter is mainly composed of *C. kawakamii* leaf and branch litter. We brought the samples back to the laboratory for mixing, picking out the leaf and branch litter prior to cleaning with distilled water. The litter was then dried to a constant weight in an oven at 65 °C for 48 h. We measured 20 g of leaf litter and 30 g of branch litter to be placed in a litter bag with dimensions of 20 cm $\times$ 20 cm and a 0.5 mm aperture at both sides. The mesh litter bags can exclude earthworms from the litter [30]. The litter volume accounted for approximately 50% of the decomposition bag.

In May 2018, we selected five plots in the east, south, west, north, and center positions from which we removed plants and litter from the soil surface in canopy gaps and non-gaps. We then randomly laid the litter bags on the soil surface. We collected the litter decomposition bags once from each site every three months, namely, the early stage (90 days), the medium stage (180 days), the late stage (270 days), and the last stage (360 days). On each plot, six litter bags of leaf and branch litter were placed, two of which were spares. Each litter bag was more than 2 cm apart. We separately placed sixty leaf and branch litter bags in each canopy gap for a total of 720 litter bags (12 canopy gaps and non-gaps $\times$ 5 positions $\times$ 2 litter types $\times$ 6 litter bags) across the whole experiment.

We placed an air temperature and humidity recorder (DS1923–F50, Maxim Integrated, San Jose, CA, USA) at the center of the canopy gap, 1.5 m from the ground and at a soil depth of 10 cm, to automatically record the air temperature, relative air humidity, and soil temperature data every 2 h. Since the air temperature, relative air humidity, and soil temperature are continuous data, we took the average air temperature, average relative air humidity, and average soil temperature to perform analyses for the four stages of litter decomposition.

### 2.3. Survey of Plant Communities

At the early stage of litter placement, we tagged and investigated all plants with a breast-height diameter of above 1 cm (trees and shrubs) in canopy gaps and non-gaps and identified the species names. Based on the investigated data, we calculated the taxonomic diversity, including the tree species richness, Shannon–Wiener index, Pielou index, Margalef richness index, and Simpson index in canopy gaps and non-gaps [31].

### 2.4. Litter Mass Remaining and Nutrient Release

For each recovery stage, we collected one leaf litter bag and one branch litter bag from each plot in different canopy gaps and non-gaps. We then shook off the impurities, such as soil blocks and fine roots adhered to the surface of the litter bags, and placed the samples into sterilized, sealed polyethylene bags before bringing them back to the lab. We used deionized water to carefully clean the impurities or soil from the surface of the leaf and branch litter and picked out the plant roots. We then packed them in marked envelopes for drying in an oven at 65 °C to a constant weight for further measurements. We crushed the constant-weight leaf and branch litter through 100 mesh sieves using a plant crusher and measured the total carbon contents (TC) and total nitrogen (TN) contents of the litter using a carbon and nitrogen analyzer (VARIO MAX CN Elemental Analyzer, Elementar Analysensysteme GmbH, Langensel–bold, Germany).

The litter mass remaining was assumed to follow the formula [32]:

$$X_t = X_0 \, e^{-kt} \tag{1}$$

where $X_t$ is the mass remaining (%) of litter decomposition $t$, $X_0$ is the initial mass of the litter, $e$ is the bottom of the natural logarithm, $t$ is the decomposition time (months), and $k$ is the decomposition coefficient of the litter (g g$^{-1}$). The greater the $k$ value, the faster the decomposition rate.

The decomposition time of the litter was modeled using the following formula [33]:

$$50\% \text{ decomposing time } (T_{50\%}) = -\ln(1 - 0.50)/k, \tag{2}$$

$$95\% \text{ decomposing time } (T_{95\%}) = -\ln(1 - 0.95)/k, \tag{3}$$

where $T_{50\%}$ indicates the time required for the litter decomposition mass remaining to be 50% and $T_{95\%}$ indicates the time required for the litter decomposition mass remaining to be 5%.

The litter nutrient return was assumed to follow the formula:

$$N = N_t - N_{t-1}, \tag{4}$$

where $N_t$ is the nutrient contents of the litter at time $t$, $N_{t-1}$ is the nutrient contents of the litter at time $(t-1)$, and $t$ is the decomposition time (months). When $N > 0$, it indicates that the litter nutrient contents express an enrichment condition, whereas when $N < 0$ it indicates that the litter nutrient contents express a release condition [33].

### 2.5. Soil Sampling and Measurement

In August 2018 and January 2019, we collected soil samples from the 0–10 cm soil layer of five plots and mixed them evenly in sterilization bags after collecting the litter

bags of each canopy gap and non-gap. The soil samples were transported back to the lab immediately, sieved through 2 mm mesh, and subdivided into two subsamples. One subsample was used to determine the physical and chemical properties of the soil, and another subsample was used to measure the microbial diversity.

The determination of the physical and chemical properties of the soil, including the soil pH, was determined by the potentiometric method (water–soil ratio 2.5:1); the soil water content was measured using the drying method; the total phosphorus content and total potassium content were determined using the sulfuric acid–perchloric acid digestion method and an inductively coupled plasma emission spectrometer (ICP–OES, PEOPTIMA 8000, PerkinElmer Inc., Waltham, MA, USA); and the total carbon content and total nitrogen content were measured using carbon and nitrogen analyzers (VARIO MAX CN Elemental Analyzer, ElementarAnalysen systeme GmbH, Langenselbold, Germany). We used the Biolog–Eco microplate method to incubate soil microorganisms for 144 h and measured the average well color development (AWCD) every 24 h, thus determining the Simpson index, Shannon–Wiener index, Pielou index, Brillouin index, and McIntosh diversity index according to the average well color development.

*2.6. Statistical Analyses*

To verify the first hypothesis, we first tested all data for normality using the Kolmogorov–Smirnov test ($\alpha$ = 0.05). We conducted Levene's test ($\alpha$ = 0.05) to assess the homogeneity of variance across the treatments. The variation in the mass remaining and nutrient release among model results was tested using a one-way ANOVA. Following the ANOVA, Tukey's multiple comparisons test was used to identify the effects of canopy gaps and decomposition stages and their interaction on the mass remaining and nutrient release of leaf and branch litter. We also fitted a litter decomposition rate curve based on the Olson model and predicted the time required for the litter decomposition mass remaining to drop to 50% and 95%.

To verify the second hypothesis, we took the environmental factors (including canopy openness, soil pH, soil water content, average soil temperature, average air temperature and relative air humidity, total potassium content, total nitrogen content, and total phosphorus content), taxonomic diversity indices (including the plant species richness index, Shannon–Wiener index, Pielou index, Margalef index, and Simpson index), and soil microbial diversity indices (including the Simpson index, Shannon–Wiener index, Pielou index, Brillouin index, and McIntosh index) as explanatory factors to assess the goodness of fit. Since the number of samples was less than the number of explanatory factors, we used a random forest analysis to filter the explanatory factors for subsequent analysis. Meanwhile, we sorted and selected the environmental factors according to their importance [34]. A random forest analysis was conducted to filter the top 10 factors of relative importance using the "random forest" package. At the early decomposition stage, this included environmental factors, such as canopy openness, soil pH, soil water content, average air temperature and relative air humidity, total potassium content, and total nitrogen content; measures of taxonomic diversity, such as the Shannon–Wiener index and Margalef index; and measures of soil microbial diversity, such as the Brillouin index. At the late decomposition stage, environmental factors included the canopy openness, soil pH, soil water content, average air temperature and humidity, total potassium contents, and total nitrogen contents; the taxonomic diversity included the Pielou index; and the soil microbial diversity included the Simpson index and Brillouin index. Then, we conducted a hierarchical analysis using the "hier.part" package to separate out the multivariate data contributions of each categorical variable and calculated the interpretation of environmental factors, taxonomic diversity, and soil microbial diversity at the early and late decomposition stages of leaf and branch litter.

A generalized linear model (GLM) was used to reveal the relationships among the nutrient release effect of the leaf and branch litter and the explanatory factors (the top 10 factors of relative importance) of canopy gaps and non-gaps. We then used the variance

expansion factor (VIF) to determine the collinearity of the explanatory factors. According to Akaike's information criterion (AIC), we optimized the model using the stepwise regression method until all variables were less than 10. Using the "broom" package, we performed a GLM analysis to analyze the primary factors and their effects influencing the enrichment and release of total carbon and nitrogen from the litter at different stages of decomposition. Since non-gaps are not completely closed stand types, their size cannot be assigned; we therefore used canopy openness instead of gap size in the correlation analysis ($R^2$ = 0.889) [35]. The data analysis was conducted in R 4.0.3 [36].

## 3. Results

### 3.1. The Environmental, Taxonomic, and Soil Microbial Diversity of Canopy Gaps

The environmental factors and taxonomic diversity between canopy gaps and non-gaps demonstrated differences, and the soil microbial activity in the early decomposition stage was higher than the soil microbial activity in the late decomposition stage (Table A2).

### 3.2. Asynchronous Decomposition of Litter in Canopy Gaps

The two-way ANOVA indicated that gap size significantly affected the litter mass remaining for leaf and branch litter (Table A3) ($p < 0.05$). At the early decomposition stage, the mass remaining for leaf litter in medium gaps was higher than in the other gaps, while it was the lowest at the last stage of decomposition (Table 1). Across the whole period of decomposition, the mass remaining for branch litter in non-gaps was higher than in other gaps.

**Table 1.** Litter mass remaining for leaf and branch in canopy gaps and non-gaps.

| Litter Types | Gap Sizes | Decomposition Time/day | | | | |
| --- | --- | --- | --- | --- | --- | --- |
| | | 0 days | 90 days | 180 days | 270 days | 360 days |
| Leaf litter | Large gaps | 20.00 a | 16.91 ± 0.16 a | 14.84 ± 0.07 a | 13.52 ± 0.81 a | 11.26 ± 0.63 a |
| | Medium gaps | 20.00 a | 17.61 ± 0.09 a | 15.33 ± 0.19 a | 12.37 ± 0.04 b | 10.15 ± 0.14 a |
| | Small gaps | 20.00 a | 17.11 ± 0.42 a | 14.62 ± 0.51 a | 12.61 ± 0.25 a,b | 10.47 ± 0.47 a |
| | Non-gaps | 20.00 a | 17.56 ± 0.65 a | 14.99 ± 0.63 a | 12.02 ± 0.70 b | 10.88 ± 0.92 a |
| | *F* | / | 1.864 | 1.244 | 4.209 | 1.556 |
| | *p* | / | 0.200 | 0.345 | 0.036 * | 0.261 |
| Branch litter | Small gaps | 30.00 a | 27.06 ± 0.54 b | 24.74 ± 0.41 a,b | 22.30 ± 0.32 a | 19.28 ± 0.61 a |
| | Medium gaps | 30.00 a | 27.25 ± 0.24 b | 23.35 ± 0.61 c | 20.72 ± 0.38 b | 17.29 ± 0.123 b |
| | Small gaps | 30.00 a | 27.10 ± 0.54 b | 24.55 ± 1.16 b,c | 21.81 ± 1.10 a,b | 18.45 ± 0.54 a,b |
| | Non-gaps | 30.00 a | 28.22 ± 0.31 a | 25.88 ± 0.63 a | 22.47 ± 0.63 a | 19.76 ± 1.00 a |
| | *F* | / | 7.630 | 7.732 | 4.656 | 7.664 |
| | *p* | / | 0.006 ** | 0.006 ** | 0.028 * | 0.006 ** |

Notes: different letters in the same column indicate the significant differences in mass remaining of leaf, and branch litter in different sizes of canopy gaps at the same decomposition stage; *p* indicates probability value; *F* indicates *F* test value. the * as $p < 0.05$, ** as $p < 0.01$.

### 3.3. Litter Decomposition Model in Canopy Gaps

The correlation coefficient $R^2$ of the litter decomposition in canopy gaps and non-gaps indicated that the Olson index model fit the decomposition of leaf and branch litter well. (Figure 1). The decomposition index *k* of leaf or branch litter in canopy gaps and non-gaps was the same, and the decomposition rate of leaf litter was higher than that of branch litter (Table 2).

### 3.4. Dynamic Changes in Total Carbon and Nitrogen Contents during Litter Decomposition

The total carbon content of leaf and branch litter in different gaps presented a net release pattern (Figure 2, Table A4) which showed a sharp downward trend at the final stage of decomposition. In the late decomposition period, the total carbon content of leaf

litter in medium gaps was higher than in other gaps, whereas there was no significant difference in the total carbon content of branch litter between gaps and non-gaps.

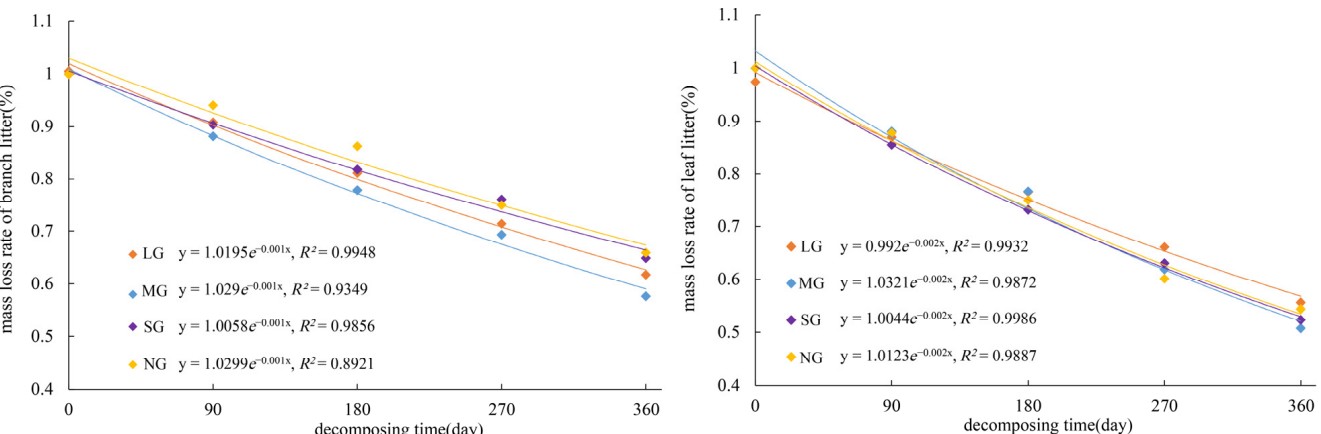

**Figure 1.** Fitting model of litter decomposition index in canopy gaps and non-gaps.

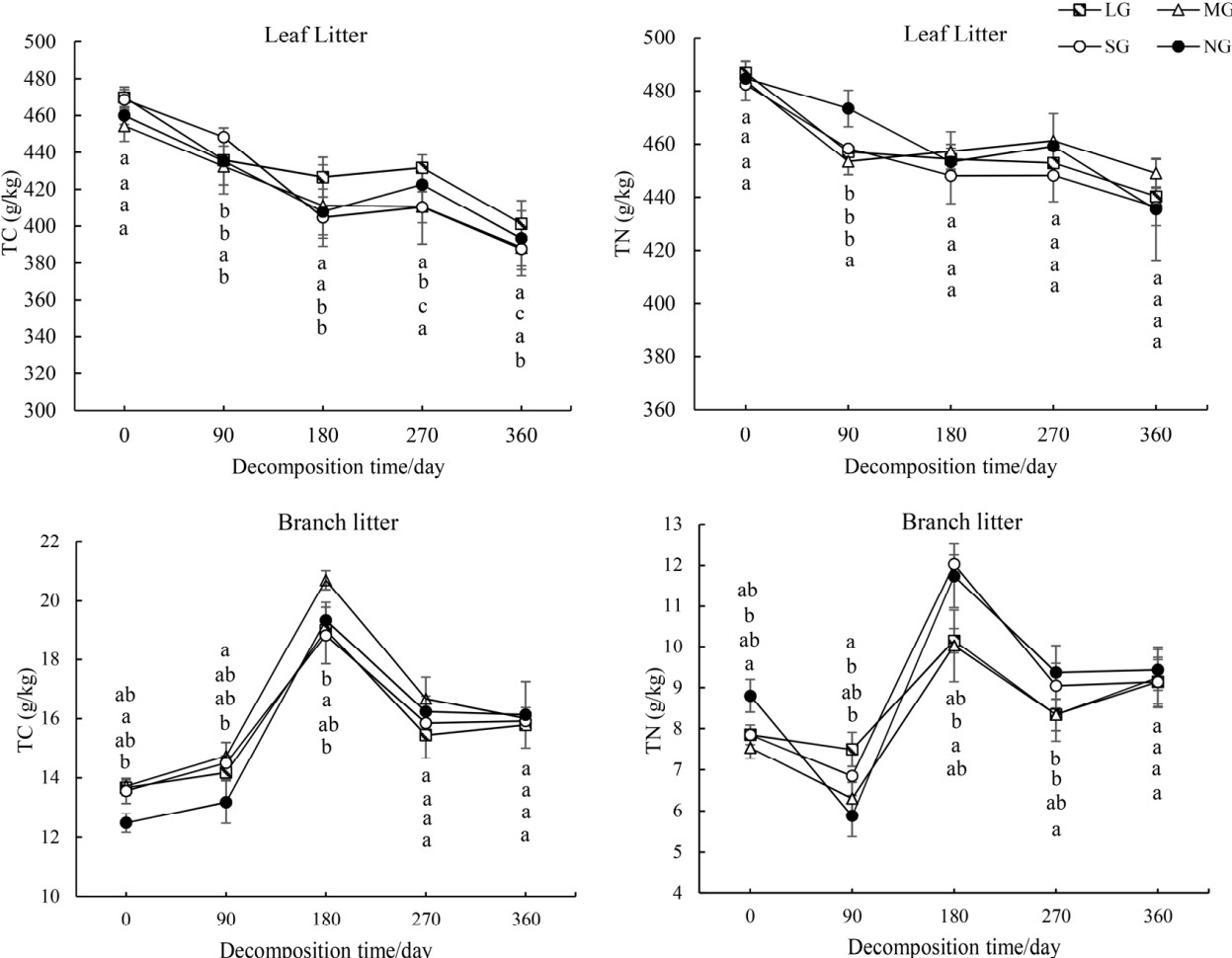

**Figure 2.** Dynamic changes in carbon and nitrogen contents of the litter in canopy gaps and non-gaps. Notes: From top to bottom in each figure, different lowercase letters in the same column indicate significant differences among large gaps, medium gaps, small gaps, and non-gaps ($p < 0.05$).

**Table 2.** Regression equation of litter mass remaining during litter decomposition.

| Litter Types | Gap Sizes | Decomposition Index ($k$) | $T_{0.5}$/day | $T_{0.95}$/day |
|---|---|---|---|---|
| Leaf litter | Large gaps | 0.002 | 342.558 | 1493.850 |
| | Medium gaps | 0.002 | 362.371 | 1513.615 |
| | Small gaps | 0.002 | 348.769 | 1500.061 |
| | Non-gaps | 0.002 | 352.686 | 1503.979 |
| Branch litter | Large gaps | 0.001 | 712.460 | 3015.045 |
| | Medium gaps | 0.001 | 721.735 | 3024.320 |
| | Small gaps | 0.001 | 698.930 | 3001.516 |
| | Non-gaps | 0.001 | 722.609 | 3025.194 |

The total nitrogen content of the leaf litter in different gaps demonstrated a single-peak enrichment–release pattern. The total nitrogen content of the branch litter showed a fluctuating pattern in the form of release–enrichment–release–enrichment (Figure 2, Table A5). During the whole decomposition period, the total nitrogen content in the leaf litter was slightly higher than in the branch litter.

### 3.5. The Importance of Driving Factors for Nutrient Release

At the early decomposition stage, the importance of environmental factors on the release of the total carbon and nitrogen contents in the leaf and branch litter was higher than that of taxonomic diversity and soil microbial diversity (Figure 3, Table A7). At the late decomposition stage, environmental factors were the main factors influencing the release of the total carbon and nitrogen contents in the leaf and branch litter (Figure 3, Table A8). For the release of carbon in branch litter, the role of soil microorganisms is stronger than the role of environmental factors.

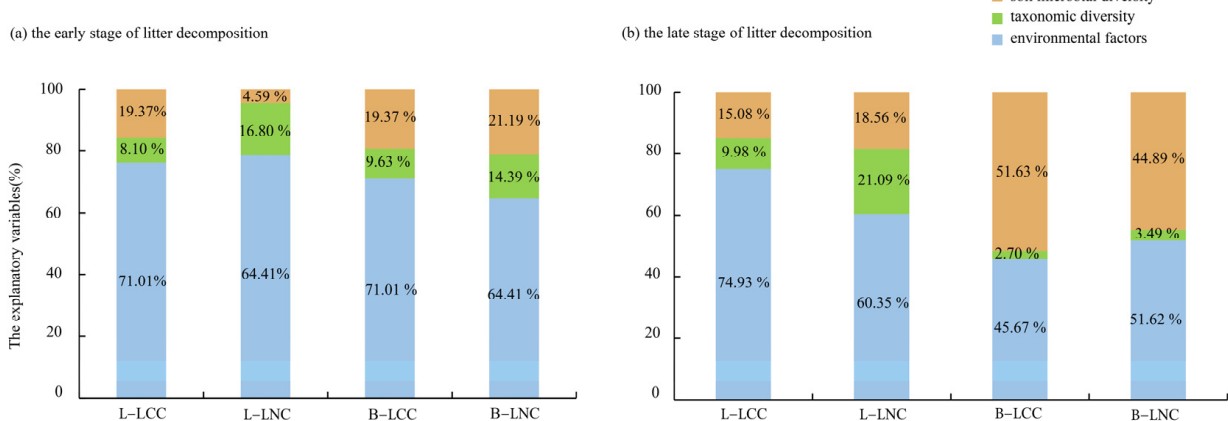

**Figure 3.** The explanatory variables of factors on litter carbon and nitrogen release in different decomposition stages of litter in canopy gaps. Notes: L−LCC indicates leaf litter carbon content; L−LNC indicates leaf litter nitrogen content; B−LCC indicates branch litter carbon content; B−LNC indicates branch litter nitrogen content.

The generalized linear model indicated (Figure 4) that the soil water content and average relative air humidity at the early decomposition stage were the main factors affecting the total nitrogen enrichment and total carbon release of the leaf litter. The soil water content, pH value, canopy openness, and the taxonomic Pielou index at the late decomposition stage were the main factors affecting the total nitrogen release. The soil total nitrogen contents, total potassium contents, soil water content, average air temperature and relative air humidity, and the Brillouin index of soil microbial diversity were the main factors affecting total carbon enrichment.

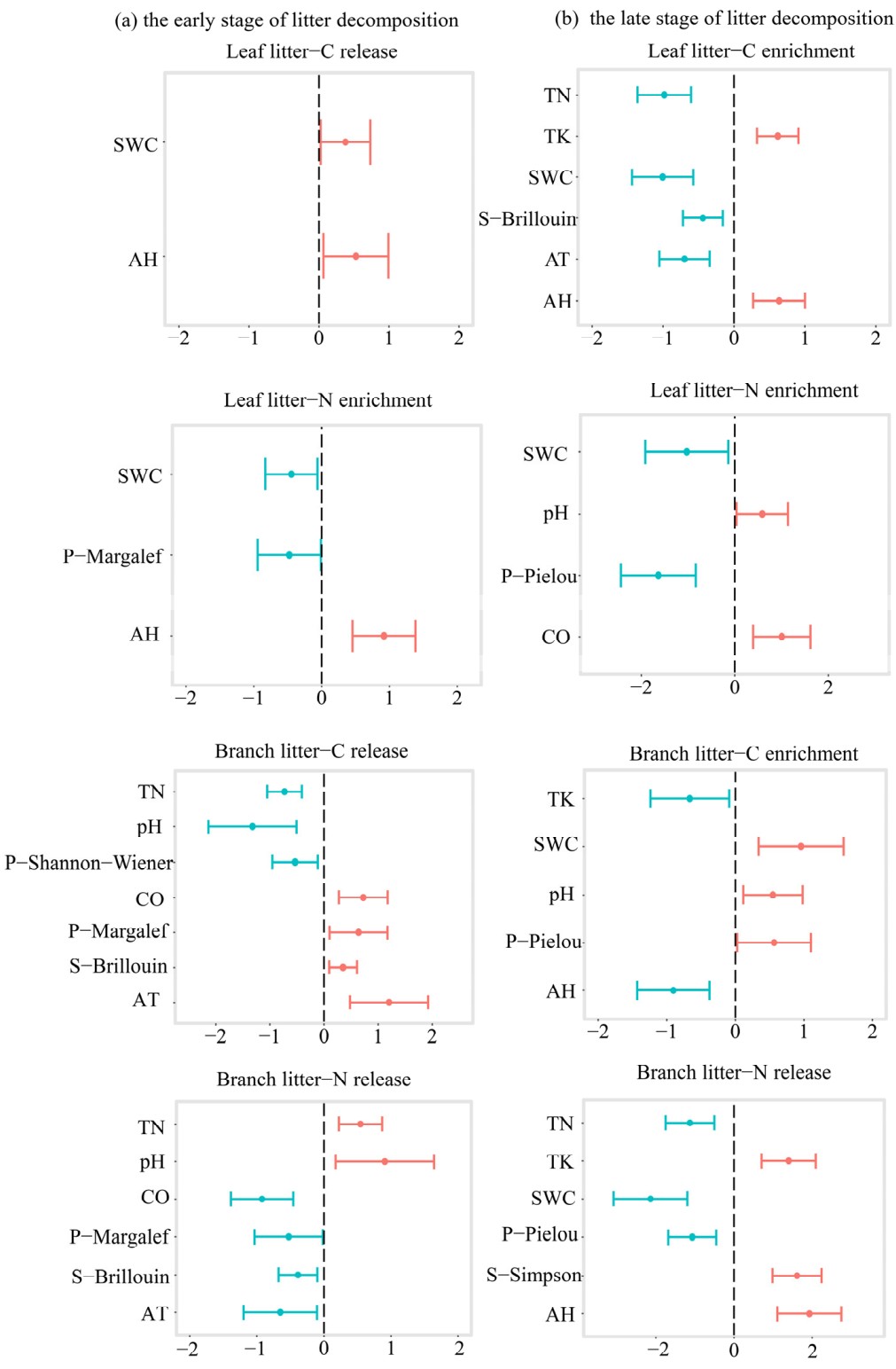

**Figure 4.** Generalized linear model analysis of the relationship among explanatory factors in canopy gaps and carbon and nitrogen contents of leaf and branch litter at different stages of litter decomposition. Notes: CO, AT, AH, TK, TN, and SWC indicate canopy openness, average air temperature, average relative air humidity, total potassium contents, total nitrogen contents of soil, and soil water content, respectively. S−Brillouin, S−Simpson indicate soil microbial Brillouin index and Simpson index, respectively. P−Shannon−Wiener, P−Pielou, and P−Margalef indicate taxonomic Shannon−Wiener index, Pielou index, and Margalef index, respectively. Horizontal coordinates represent 95% confidence intervals; red lines indicate positive effect; blue lines indicate negative effect.

At the early stage of decomposition, the total carbon and nitrogen contents of the branch litter were significantly correlated with the average air temperature, soil total nitrogen contents, pH value, canopy openness, the taxonomic Margalef index, and the Brillouin index of soil microbial diversity (Figure 4). Canopy openness, average air temperature, the taxonomic Margalef index, and the Brillouin index of soil microbial diversity had a significant negative correlation with the total nitrogen release and a positive correlation with the total carbon release. At the late stage of decomposition, the soil total potassium content, average relative air humidity, and Simpson index of the soil microbial diversity had a significantly positive correlation with the total nitrogen release, and the soil total nitrogen content, soil water content, and taxonomic Pielou index had a negative correlation. The soil total potassium content and the average relative air humidity had a negative correlation with the total carbon enrichment of the branch litter.

## 4. Discussion

### 4.1. Characteristics of Litter Mass Remaining in Canopy Gaps and Non-Gaps

Canopy gaps are an important driving force for regulating the material circulation of forest ecosystems, influencing the feedback mechanism of aboveground plant–litter–soil interactions [21]. At the early stage of decomposition, the decomposition rate of the leaf and branch litter in large gaps was the fastest, and the litter mass remaining was at its lowest. However, in the last stage of decomposition, the litter mass remaining in large gaps was the greatest, and the decomposition of leaf and branch litter in medium gaps was the slowest. This may be because the litter decomposition is asynchronous. With an increase in the canopy gap size, the center position in large gaps receives stronger effective photosynthetic radiation and precipitation, enhancing the leaching and photodegradation of understory plants and thus accelerating the litter decomposition rate at the early stage of decomposition [22]. Furthermore, canopy gaps promote taxonomic diversity and stimulate microbial enzyme activity by secreting organic substances from roots, thus accelerating litter decomposition [14,15]. Meanwhile, at the last stage of decomposition, the soil microbial community may become the main limiting factor for litter decomposition. This is likely due to the serious fragmentation of litter, the reduction in easily decomposed substances by the end of decomposition, and the decrease in the strength of hydrothermal conditions on recalcitrant substances. A higher soil fungal diversity of medium gaps in the forest promotes the enrichment of enzyme activity, stimulates the litter decomposition, and alters the litter mass remaining [21], although certain macroinvertebrates were excluded from the litter bags, lowering the rates of litter comminution. Litter bags can be used experimentally to examine decomposition rates at various time scales and the contribution of different factors (e.g., environmental factors, taxonomic diversity, and soil microbial diversity) [25], contributing to understanding the feedback mechanism of litter–taxonomic diversity–soil microbial community.

After the whole experiment, the remaining mass of leaf and branch litter in the medium gaps was lower than that in other canopy gaps. However, we predicted that the time required for the decomposition of 95% of the litter mass in the medium canopy gaps was not completely less than that in other canopy gaps in the Olson model. This may be due to the change in the litter state and the change in the litter decomposition rate due to environmental factors in different canopy gaps at different stages of decomposition. For example, the soil total phosphorus content changed with the change in decomposition time, thus influencing the soil C/P ratio, and the change in rhizosphere soil properties had a feedback regulation effect on plant growth, which affected the nutrient absorption by the plant roots. In addition, the soil environment indirectly affected plant growth through the nutrient mineralization of the soil microbial diversity [7]. In turn, the metabolites secreted by plant roots improved the soil nutrients and affected the soil microbial activity [7,37]. The plant and soil feedback promotes the soil carbon cycle [7], thereby affecting the litter decomposition process. The decomposition rate of the branch litter in canopy gaps was significantly lower than that of leaf litter, which may be related to the chemical charac-

teristics of the litter. Chen found that the C/N ratio was an important indicator of litter decomposition rate [38], with high litter C/N ratios being associated with high lignin concentrations, which can, in turn, reduce decomposition rates [39]. Our results showed that the C/N ratio of the branch litter was higher than that of the leaf litter (Table A6), further verifying the existence of chemical differences in litter decomposition rates.

### 4.2. Total Carbon Contents during Litter Decomposition in Canopy Gaps and Non-Gaps

Litter decomposition determines the carbon return and the nutrient cycle of the ecosystem, which is a key part of the carbon balance in terrestrial ecosystems [1,40]. The environmental conditions present inside and outside the litter bags are similar [23]. However, the microenvironment can be altered with the formation of canopy gaps, which can have a direct or indirect influence on the nutrient release process from the leaf and branch litter inside the litter bags [1,41]. The total carbon release rate at the early decomposition stage was higher than that at the late decomposition stage, indicating that the litter carbon release was asynchronous. It may be that the rapid leaching of soluble carbohydrates in the early decomposition stage led to a high total carbon release rate. In the late decomposition stage, the litter was affected by the external environment or exogenous nutrient input and other factors, which indirectly affected the litter decomposition via changing the microbial activity and community composition by detritivore arthropods [41,42]. At the early stage of decomposition, the total carbon release of the branch litter in large gaps was higher than in other gaps (Figure 2). This could be due to the presence of high taxonomic diversity in the large gaps, which affected the microbial abundance and community structure and tended to result in synergistic or antagonistic interactions [43–45]. Meanwhile, large gaps also experienced strong ultraviolet radiation, which may have accelerated the carbon turnover efficiency in such gaps [46].

At the late decomposition stage, the total carbon contents of the leaf and branch litter in different canopy gaps demonstrated a state of micro-enrichment (Figure 2). This may be due to the nutrients released from the decomposition of plant metabolites being predominantly used for storage in the winter, with the growth rate slowing down or withering, resulting in the enrichment of the total carbon in the litter. The total carbon release of the branch litter in large gaps was stronger than in other gaps. This may be due to the higher soil pH value in the large gaps increasing the abundance and diversity of soil animals, thus accelerating litter decomposition through fragmentation and enhanced microbial colonization [46,47]. This viewpoint was also confirmed by the significant positive effect of the soil pH value on the total carbon enrichment of the branch litter (Figure 4). Meanwhile, the increase in the soil pH value inputs energy-rich, available carbon through the plant rhizosphere excitation effect, enhances the soil's microbial enzyme secretion and organic matter mineralization, and further promotes the whole process of carbon release [48].

The total carbon release at the early decomposition stage in large gaps was higher than in other gaps, which may be due to the ability of large gaps to regulate the water and heat exchange conditions of the forest canopy and understory, enhance the leaching effect of rainfall on litter, and indirectly improve the turnover efficiency and nutrient release [49]. Furthermore, there was a significant positive correlation between taxonomic diversity and litter nutrient release (Figure 4); therefore, the high taxonomic diversity in large gaps directly affected the litter type and chemical composition, resulting in shifts in the microbial communities and litter decomposition rate. In particular, the high chemical diversity of the litter provided a diverse diet for decomposers, which have a strong impact on microbial communities and functional diversity and hence increase the decomposition rate, providing positive feedback to plants [50,51]. The canopy shade produced by small gaps and non-gaps reduced the cumulative irradiance and exposure to sunlight [52], thereby reducing the total carbon release process of the litter. At the late decomposition stage, the total carbon enrichment capacity of the litter in non-gaps was strong, potentially due to the high soil total potassium content in non-gaps promoting the soil microbial activity and functional diversity [21] and directly affecting the litter oxidation process. It was previously

proved that the soil microbial communities regulate the litter's total carbon and nitrogen release in the early decomposition stages, whereas the soil water content plays a major role in the late decomposition stage [41]. Our results also confirmed this perspective on the differences in the primary factors affecting the nutrient release of branch litter at different stages of decomposition.

*4.3. Total Nitrogen Contents during Litter Decomposition in Canopy Gaps and Non-Gaps*

Nitrogen is the most important element in plants, being critical to the growth, development, and reproduction of soil microorganisms. The enrichment and release of the total nitrogen in litter are key links that affect the soil nutrient cycle and maintain the stability of forest ecosystems [53]. When the C/N ratios of different types of forest litter were higher than 5–15, it indicated that the total nitrogen in the litter was enriched; when these ratios were lower than 5–15, the nitrogen in the litter began to be released [54]. The C/N ratios of the leaf and branch litter in *C. kawakamii* canopy gaps were higher than 15. The total nitrogen contents of the leaf litter presented a single peak model of enrichment–release (Figure 4), while the total nitrogen contents of the branch litter showed a fluctuation in the form of release–enrichment–release–enrichment, indicating that the total nitrogen release and the enrichment of litter was asynchronous. This may be because the total nitrogen release from litter in a subtropical evergreen broad-leaved forest is not determined by the C/N ratio [41]. Therefore, we might need to consider the comprehensive effect of environmental factors.

At the early decomposition stage, the total nitrogen enrichment rate of the leaf litter in large gaps was lower than in the other gaps. This may be due to the reduction in the interception effect on precipitation in large gaps in addition to the fast release of soil nitrogen [22,23]. A change in the soil nitrogen content alters the resources available to microbes and drives microbial activity and community structure, in turn affecting processes of decomposition [55]. At the late decomposition stage, the total nitrogen release of the litter in medium gaps was slightly stronger than in the other gaps, which may be due to the high soil water content in canopy gaps hindering the oxygen supply for soil microorganisms and inhibiting soil microbial activity by restricting soil aeration [46]. Excessive soil water content also affects the number, size, and branching pattern of plant roots, alters the physiological and ecological processes of plant roots, and regulates the growth and development of plants through the feedback of soil properties, which indirectly affects the litter decomposition [56]. At the same time, the high canopy density and low diversity of understory plants in non-gaps and small gaps led to a decrease in the heterogeneity of litter resources. This may have affected the food sources for soil animals and microorganisms and caused the effective niche to adapt to the aggregation of various microorganisms, leading to a reduction in the utilization efficiency of soil nutrients and thus affecting the total nitrogen release of litter [20,57]. In addition, non-gaps and small gaps had a high canopy density, and the leaching effect of the understory vegetation was weakened. The appropriate hydrothermal conditions in the medium gaps improved the functional diversity of soil microorganisms and ensured a sufficient supply of carbon sources in the forest soil, which increased the activity of extracellular enzymes and promoted the type and quantity of the organic substances secreted by them, thus promoting the litter nutrient release process [52,58]. This demonstrates that gap sizes affect soil microorganisms and plant communities through environmental factors such as precipitation, subsequently affecting the total nitrogen release of the litter.

At the early stage of decomposition, the total nitrogen release of the branch litter in non-gaps was strong, while it was relatively weak in canopy gaps. This could be due to the rapid mineralization of microbial nitrogen into nitrogen molecules with a high mobility, promoting the balance of nutrients during the process of soil microbial proliferation and reproduction [58]. Therefore, the nitrogen-induced increase in soil fauna density may stimulate fauna-driven litter decomposition. In addition, with the increase in gap size, the strong light, including ultraviolet light, led to the rapid warming of the soil [52]. A

high soil temperature promoted the mineralization ability of the soil total nitrogen content, improved the availability of soil nutrients, and reduced the input of soil carbon and root exudates to a certain extent, resulting in a decrease in the abundance and activity of plant inter-rooted soil microorganisms. This notion is also supported by the high soil nitrogen content found in non-gaps. Meanwhile, soil microorganisms preferred to use root exudates, thereby inhibiting the litter's release of total nitrogen. At the same time, a high soil temperature inhibited soil respiration and microbial activity [59], and the variation in the soil environment indirectly affected the nutrient release process of the litter. At the late decomposition stage, the total nitrogen release of the branch litter in small gaps was the strongest, which might be due to the decrease in soil enzyme activity in the winter. Low soil enzyme activity directly affects the activity of soil microorganisms, which indirectly affects plant root secretion and reduces the degradation ability of soil microorganisms in litter [59], further confirming that plant–litter–soil feedback regulates forest nitrogen cycling and indirectly affects the stability of forest ecosystems.

## 5. Conclusions

Litter decomposition and the total release and enrichment of carbon and nitrogen for leaf and branch litter are asynchronous. This is embodied in the difference in the litter decomposition rates, the variation in the total carbon and nitrogen contents, and the differences in the release or enrichment rates during different stages of decomposition. Environmental factors and the soil microbial community play vital roles in litter decomposition and nutrient release due to the formation of canopy gaps. We conclude that canopy gaps control the litter decomposition and nutrient release in subtropical forests, which will deepen our understanding of soil ecological balance and nutrient cycling.

**Author Contributions:** Conceptualization, J.C.; methodology, J.C. and J.Z.; software, J.C.; validation, J.L.; formal analysis, Z.W. and J.C.; investigation, J.C., C.X., J.Z., X.W. and C.W.; data curation, J.Z. and Z.W.; writing—original draft preparation, J.C.; writing—review and editing, J.C. and D.X.; visualization, J.C., J.Z., B.C. and X.W.; supervision, J.L., D.X. and Z.H.; funding acquisition, J.L. and Z.H. All authors have read and agreed to the published version of the manuscript.

**Funding:** This work was supported by the National Natural Science Foundation of China (grant numbers 31700550, 31770678); the Science and Technology Promotion of Project Forestry Bureau of the Fujian Province (grant number 2022FKJ21); and Forestry Peak Discipline Construction Project of Fujian Agriculture and Forestry University (grant number 72202200205).

**Institutional Review Board Statement:** Not applicable.

**Informed Consent Statement:** Not applicable.

**Data Availability Statement:** Not applicable.

**Acknowledgments:** We wish to express our thanks for the support received from the *Castanopsis kawakamii* Nature Reserve in Sanming City, Fujian Province, for allowing us to collect samples. The authors wish to thank Xinguang Gu, Mengjia Li, Lan Jiang, Jingyu Xiao, and Qingrong Huang for the experimental work. The authors also sincerely appreciate the helpful and constructive comments provided by the reviewers of the draft manuscript.

**Conflicts of Interest:** The authors declare no conflict of interest.

# Appendix A

**Table A1.** General information on canopy gaps and non-gaps in the natural forest of *C. kawakamii*.

| Gap Size | Cause of Gap Formation | Size/m$^2$ | Elevation/m | Canopy Openness/% | Ratio of Crow Inclination/% | Aspect | Slope Position | Slope/° |
|---|---|---|---|---|---|---|---|---|
| Large gaps | Tree fall | 210.56 | 224 | 23.93 | 27.25 | Southeast | Mid-slope | 22 |
| | Tree fall | 200.38 | 211 | 31.96 | 33.69 | Southeast | Mid-slope | 31 |
| | Natural death/trunk broken | 207.57 | 214 | 29.91 | 31.81 | Southeast | Upper-slope | 29 |
| Medium gaps | Trunk broken | 74.22 | 196 | 11.56 | 36.39 | Southeast | Down-slope | 28 |
| | Tree fall | 50.59 | 214 | 17.33 | 35.47 | Northeast | Upper-slope | 31 |
| | Tree fall | 74.65 | 188 | 19.21 | 26.84 | Southwest | Down-slope | 29 |
| Small gaps | Branch broken | 32.11 | 225 | 5.67 | 31.51 | Northwest | Upper-slope | 14 |
| | Branch broken | 31.59 | 203 | 7.46 | 39.68 | Southwest | Mid-slope | 11 |
| | Branch broken | 36.78 | 214 | 8.32 | 38.43 | Northwest | Down-slope | 28 |
| Non-gaps | / | 100 | 221 | 5.62 | / | West | Down-slope | 13 |
| | / | 100 | 221 | 5.47 | / | West | Down-slope | 13 |
| | / | 100 | 221 | 5.53 | / | West | Down-slope | 13 |

**Table A2.** Explanatory factors for litter decomposition in canopy gaps and non-gaps.

| Explanatory Factors | | The Early Stage of Litter Decomposition | | | The Late Stage of Litter Decomposition | | |
|---|---|---|---|---|---|---|---|
| | | Maximum Value | Minimum Value | Coefficient of Variation/% | Maximum Value | Minimum Value | Coefficient of Variation/% |
| Environmental factors | Canopy openness | 31.96 | 5.47 | 69.02 | 31.96 | 5.47 | 69.02 |
| | ST/°C | 26.35 | 24.92 | 1.67 | 13.36 | 10.61 | 5.69 |
| | AT/°C | 27.23 | 25.86 | 1.58 | 12.36 | 11.19 | 3.75 |
| | AH/% | 96.86 | 90.17 | 2.10 | 100.35 | 95.97 | 1.54 |
| | Soil pH | 3.49 | 3.17 | 2.81 | 3.59 | 3.30 | 2.87 |
| | SWC/% | 30.68 | 24.70 | 6.15 | 29.62 | 22.90 | 6.52 |
| | Soil TN/g·kg$^{-1}$ | 32.49 | 23.16 | 11.5 | 33.33 | 17.89 | 14.70 |
| | Soil TP/g·kg$^{-1}$ | 0.18 | 0.10 | 15.78 | 0.20 | 0.13 | 12.58 |
| | Soil TK/g·kg$^{-1}$ | 42.30 | 29.29 | 9.85 | 38.14 | 25.32 | 11.09 |
| Taxonomic diversity | Shannon-Wiener index | 3.92 | 2.35 | 20.02 | 3.92 | 2.35 | 20.02 |
| | Pielou index | 1.14 | 0.73 | 13.88 | 1.14 | 0.73 | 13.88 |
| | Margalef index | 7.00 | 3.08 | 27.38 | 7.00 | 3.08 | 27.38 |
| | Simpson index | 0.92 | 0.89 | 2.69 | 0.92 | 0.89 | 2.69 |
| | Species richness | 40.00 | 13.00 | 38.88 | 40.00 | 13.00 | 38.88 |
| Soil microbial diversity | Simpson index | 0.99 | 0.98 | 0.06 | 1.00 | 0.99 | 2.80 |
| | Shannon-Wiener index | 4.89 | 4.79 | 0.68 | 4.66 | 4.43 | 1.72 |
| | Pielou index | 0.99 | 0.97 | 0.68 | 0.94 | 0.89 | 1.74 |
| | Brillouin index | 3.96 | 3.83 | 1.15 | 3.47 | 2.85 | 5.34 |
| | McIntosh index | 0.95 | 0.94 | 0.20 | 1.2 | 0.96 | 6.60 |

Notes: ST indicates average soil temperature; AT indicates average air temperature; AH indicates average relative air humidity; SWC indicates soil water content. Soil TN indicates soil total nitrogen; Soil TP indicates soil total phosphorus; Soil TK indicates soil total potassium.

**Table A3.** Two-way ANOVA results for the effects of gap size and time on mass changes of leaf and branch litter.

| Factors | df | Leaf Litter | | Branch Litter | |
|---|---|---|---|---|---|
| | | *F* | *p* | *F* | *p* |
| Gap sizes | 2 | 3.418 | 0.085 | 5.816 | <0.001 *** |
| Decomposition time | 3 | 450.51 | <0.001 *** | 71.001 | <0.001 *** |
| Gap sizes × Decomposition time | 6 | 3.013 | <0.001 *** | 0.625 | 0.060 |

Notes: df indicates degree of freedom; *p* indicates probability value; *F* indicates *F* test value; *** as *p* < 0.001.

**Table A4.** Total carbon release of litter at different decomposition stages in canopy gaps and non-gaps.

| Litter Types | Gap Sizes | Decomposition Time (day) | | | |
|---|---|---|---|---|---|
| | | 90 days | 180 days | 270 days | 360 days |
| Leaf litter | Large gaps | $N < 0$ | $N < 0$ | $N > 0$ | $N < 0$ |
| | Medium gaps | $N < 0$ | $N < 0$ | $N < 0$ | $N < 0$ |
| | Small gaps | $N < 0$ | $N < 0$ | $N > 0$ | $N < 0$ |
| | Non-gaps | $N < 0$ | $N < 0$ | $N > 0$ | $N < 0$ |
| Branch litter | Large gaps | $N < 0$ | $N > 0$ | $N < 0$ | $N > 0$ |
| | Medium gaps | $N < 0$ | $N > 0$ | $N < 0$ | $N > 0$ |
| | Small gaps | $N < 0$ | $N > 0$ | $N < 0$ | $N > 0$ |
| | Non-gaps | $N < 0$ | $N > 0$ | $N < 0$ | $N > 0$ |

Notes: $N > 0$ indicates that the litter nutrient contents express an enrichment condition; $N < 0$ indicates litter nutrient contents express a release condition.

**Table A5.** Total nitrogen release of litter at different decomposition stages in different canopy gaps.

| Litter Types | Gap Sizes | Decomposition Time (day) | | | |
|---|---|---|---|---|---|
| | | 90 days | 180 days | 270 days | 360 days |
| Leaf litter | Large gaps | $N > 0$ | $N > 0$ | $N < 0$ | $N > 0$ |
| | Medium gaps | $N > 0$ | $N > 0$ | $N < 0$ | $N < 0$ |
| | Small gaps | $N > 0$ | $N > 0$ | $N < 0$ | $N > 0$ |
| | Non-gaps | $N > 0$ | $N > 0$ | $N < 0$ | $N < 0$ |
| Branch litter | Large gaps | $N < 0$ | $N < 0$ | $N < 0$ | $N < 0$ |
| | Medium gaps | $N < 0$ | $N < 0$ | $N > 0$ | $N < 0$ |
| | Small gaps | $N < 0$ | $N < 0$ | $N > 0$ | $N < 0$ |
| | Non-gaps | $N < 0$ | $N < 0$ | $N > 0$ | $N < 0$ |

**Table A6.** C/N ratio of litter in canopy gaps and non-gaps.

| Litter Types | Gap Sizes | Decomposition Time (day) | | | | |
|---|---|---|---|---|---|---|
| | | 0 days | 90 days | 180 days | 270 days | 360 days |
| Leaf litter | Large gaps | 34.23 ± 0.58 a,b | 31.88 ± 0.58 a | 22.46 ± 0.88 a | 27.95 ± 0.73 a | 25.42 ± 0.94 a |
| | Medium gaps | 33.12 ± 0.72 b | 29.35 ± 0.66 a | 19.85 ± 0.93 a | 24.63 ± 0.23 a | 24.27 ± 0.52 a |
| | Small gaps | 34.56 ± 1.46 a,b | 30.93 ± 0.70 a | 21.50 ± 0.34 a | 25.89 ± 1.05 a | 24.36 ± 0.44 a |
| | Non-gaps | 37.01 ± 0.88 a | 33.02 ± 1.34 a | 21.09 ± 0.59 a | 26.03 ± 1.08 a | 24.38 ± 2.44 a |
| | *F* | 3.210 | 2.215 | 1.655 | 0.907 | 1.559 |
| | *p* | 0.070 | 0.161 | 0.222 | 0.469 | 0.266 |
| Branch litter | Large gaps | 61.63 ± 0.16 a | 60.98 ± 2.21 a | 44.77 ± 0.84 a,b | 54.31 ± 1.61 a | 48.18 ± 3.00 a |
| | Medium gaps | 64.17 ± 3.31 a | 71.97 ± 5.44 a,b | 45.62 ± 4.57 a | 55.31 ± 5.31 a | 48.54 ± 2.67 a |
| | Small gaps | 62.46 ± 2.10 a | 66.96 ± 0.77 b | 37.25 ± 1.50 b | 49.55 ± 4.01 a | 47.73 ± 1.51 a |
| | Non-gaps | 55.07 ± 2.76 b | 80.55 ± 8.15 a | 38.60 ± 3.14 a,b | 49.02 ± 4.86 a | 46.16 ± 4.26 a |
| | *F* | 4.300 | 5.884 | 3.402 | 0.921 | 0.270 |
| | *p* | 0.044 * | 0.020 * | 0.074 | 0.473 | 0.846 |

Notes: different letters in the same column indicate the significant differences in C/N of leaf, and branch litter in different sizes of canopy gaps at the same decomposition stage; * as *p* < 0.05.

**Table A7.** The explanatory variables for factors in litter nutrient release at the early decomposition stage.

| Factors | | L−LCC/% | L−LNC/% | B−LCC/% | B−LNC/% |
|---|---|---|---|---|---|
| Environmental factors | Canopy openness | 3.45 | 5.40 | 12.16 | 27.77 |
| | Soil pH | 7.10 | 6.68 | 9.72 | 10.98 |
| | SWC | 15.14 | 8.40 | 4.25 | 1.70 |
| | Soil TN/g·kg$^{-1}$ | 1.87 | 4.02 | 16.39 | 11.42 |
| | Soil TK/g·kg$^{-1}$ | 1.82 | 14.65 | 9.63 | 1.92 |
| | AT | 14.40 | 7.19 | 16.53 | 8.60 |
| | AH | 32.26 | 32.26 | 2.33 | 2.04 |
| Taxonomic diversity | Margalef index | 3.59 | 10.68 | 5.52 | 8.64 |
| | Shannon-Wiener index | 4.50 | 6.12 | 4.10 | 5.76 |
| Soil microbial diversity | Brillouin index | 15.86 | 4.59 | 19.37 | 21.19 |

Notes: L−LCC indicates leaf litter carbon contents; L−LNC indicates leaf litter nitrogen contents; B−LCC indicates branch litter carbon contents; B−LNC indicates branch litter nitrogen contents.

**Table A8.** The explanatory variables for factors in litter nutrient release at the late decomposition stage.

| Factors | | L−LCC/% | L−LNC/% | B−LCC/% | B−LNC/% |
|---|---|---|---|---|---|
| Environmental factors | Canopy openness | 6.35 | 8.82 | 2.91 | 3.99 |
| | Soil pH | 17.40 | 8.49 | 4.13 | 9.08 |
| | SWC | 2.58 | 3.22 | 4.24 | 3.82 |
| | Soil TN/g·kg$^{-1}$ | 4.13 | 20.58 | 8.29 | 3.40 |
| | Soil TK/g·kg$^{-1}$ | 25.77 | 5.05 | 4.10 | 16.43 |
| | AT | 15.82 | 8.44 | 6.32 | 8.54 |
| | AH | 2.89 | 5.75 | 15.68 | 6.36 |
| Taxonomic diversity | Pielou index | 9.98 | 21.09 | 2.70 | 3.49 |
| Soil microbial diversity | Simpson index | 3.15 | 7.53 | 26.67 | 14.69 |
| | Brillouin index | 11.93 | 11.02 | 24.96 | 30.20 |

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
