# Peer review of "Canopy Gaps Control Litter Decomposition and Nutrient Release in Subtropical Forests"

_forests, doi:10.3390/f14040673_

Round 1
Reviewer 1 Report
The manuscript present a very interesting analysis on litter composition controlled by canopy gaps. It fills a gap in knowledge how the litter composition can be controlled by canopy, serving as a referenceThe methods and matherials are adequate. Results and discussion too. The authors should provide more photos and details about the experimental design. A schematic figure may be interesting for readers who want to reply such experiments in other areas around the world.
Reviewer 2 Report
Manuscript ID: forests-2286402
Type: Article
Title: Canopy gaps control litter decomposition and nutrient release in subtropical forests
Authors: Jiajia Chen , Jing Zhu , Ziwei Wang , Cong Xing , Bo Chen , Xuelin Wang , Chensi Wei , Jinfu Liu , Zhongsheng He , Daowei Xu *
Dear authors
The manuscript entitled "Canopy gaps control litter decomposition and nutrient release in subtropical forests" has studied the effects of canopy gaps of different sizes (three large gaps (above 150 m2), three medium gaps (50–100 m2), and three small gaps (30–50 m2) on litter decomposition and nutrient release, and reveal the main factors that influence variation during the different stages of decomposition with total of 720 litterbags (12 canopy gaps and non-gaps×5 positions×2 litter types×6 litterbags) across the whole experiment by litterbag study. Statistical analyses have done. Results showed the litter mass remaining of leaves and branches in medium gaps (50–100 m2) was lower than that of other gaps, and the leaf litter decomposition rate was higher than that of branch litter decomposition. Likewise, litter mass remaining, total carbon and nitrogen release, and enrichment were found to be asynchronous. The subject of the research work is original and has been able to provide a lot of new information in the field of mimicking nature to better management of subtropical forest of Castanopsis kawakamii.
The authors were able to answer the research questions due to the review of suitable sources, the study method of the region and the sufficient number of samples, and there is no need for additional items in the method and also other controls. The authors were able to match the research conclusions well with the evidence and arguments presented and address the main question raised. References were well presented and good previous researches were listed, however, some related references proposed that should be mentioned in the manuscript. Tables and figures are well organized and logically consistent with the content of the text.
The study design is robust and the topic fits well to the scope of the journal. The manuscript is generally clearly designed, written and illustrated. The discussion of the manuscript also well written. I recommend this paper for addressing some minor comments presented on the PDF file.
Best regards

Reviewer 3 Report
This study is about Canopy gaps control litter decomposition and nutrient release in subtropical forests. It is a relevant study for the understanding of forest functioning and will allow decisions to be made in the management of forest stands in the future.
1. What is the main question addressed by the research?
This research addresses canopy gaps control litter decomposition and nutrient release in subtropical forests. 2. Do you consider the topic original or relevant in the field? Does it address a specific gap in the field? This is a little studied topic and relevant to the knowledge of forest and plantation forest functioning. 3. What does it add to the subject area compared with other published material? This study provides novel information on the effects of canopy gaps of different sizes on litter decomposition and nutrient release, and reveals the main factors influencing variation during different stages of decomposition. 4. What specific improvements should the authors consider regarding the methodology? What further controls should be considered? In my opinion the methodology used is adequate. 5. Are the conclusions consistent with the evidence and arguments presented and do they address the main question posed? and do they address the main question posed? The conclusions are well founded and consistent. 6. Are the references appropriate? The references are appropriate.7. Please include any additional comments on the tables and figures.
Author Response
Dear editor and reviewers,
We are truly grateful to yours and reviewers’ critical comments and thoughtful suggestions for our manuscript entitled “Canopy gaps control litter decomposition and nutrient release in subtropical forests” (ID: forests- 2286402). Based on these comments and suggestions, we have made careful modifications on our manuscript. The changes based on reviewers were marked in red color. We hope the revised manuscript will meet your magazine’s standard. On behalf of all the authors of this article, I would like to express my gratitude to the reviewers for their suggestions not only to make our article better, but also to increase the breadth and depth of our research. I will answer each reviewer's questions and comments one by one.
Best regards!
Sincerely yours,